

# Signatures of red-shifted footpoints in the quiescent coronal loop system

Yamini K. Rao [1], Abhishek K. Srivastava [1], Pradeep Kayshap [2], and Bhola N. Dwivedi [1]

[1]Department of Physics, Indian Institute of Technology (BHU), Varanasi-221005, India.
[2]Institute of Physics, University of South Bohemia, Branišovská 1760, CZ – 370 05 České Budějovice, Czech Republic.

**Correspondence:** Abhishek K. Srivastava (asrivastava.app@itbhu.ac.in)

**Abstract.**

We observed quiescent coronal loops using multi-wavelength observations from the Atmospheric Imaging Assembly (AIA) onboard the Solar Dynamics Observatory (SDO) on 2016 April 13. The flows at the footpoints of such loop systems are studied using spectral data from Interface Region Imaging Spectrograph (IRIS). The Doppler velocity distributions at the footpoints lying in the moss region show the negligible or small flows at Ni I, Mg II k3 and C II line corresponding to upper photospheric and chromospheric emissions. Significant red-shifts (downflows) ranging from (1 to 7) km s$^{-1}$ are observed at Si IV (1393.78 Å; $log(T/K) = 4.8$) which is found to be consistent with the existing results regarding dynamical loop systems and moss regions. Such downflows agree well with the impulsive heating mechanism reported earlier.

## 1   Introduction

The active regions dominated by various loop structures are of significant importance for the study of coronal heating since these loop systems act as a fundamental unit of the solar corona (Klimchuk 2006; Reale 2014; De Moortel & Browning 2015). Moss region being the subset of plage regions (Fletcher & De Pontieu 1999) and transition region emission of hot core loops will provide us the better understanding about the flows and thus energy transfer mechanism between the transition region (TR) and corona.

Klimchuk (2006) has provided with a full review of the coronal heating problem. It describes that the coronal heating mechanisms are impulsive when explored from the perspective of elemental magnetic flux strands. It has also been well established that the loop structures emit significantly in the solar corona which have been classified depending on their temperatures. The spectral studies of these loop systems in response to the Doppler shift provides clue to distinguishing between the steady and impulsive heating mechanism (Del Zanna 2008; Brooks et al. 2011).

Various types of loops are hot core loops (Del Zanna, 2008), warm loops (Del Zanna et al., 2011), fan loops (Young et al. 2012; Warren & Brooks 2009 and references therein) and cool loops (Huang et al. 2015; Rao et al. 2019) present in the different regions of the solar atmosphere. The temperature and density diagnostics of quiescent coronal loops have been fairly studied earlier (Del Zanna & Mason 2003). However, there have been not much observations regarding the flows in the resolved strands/flux tubes of such loops in the solar corona.



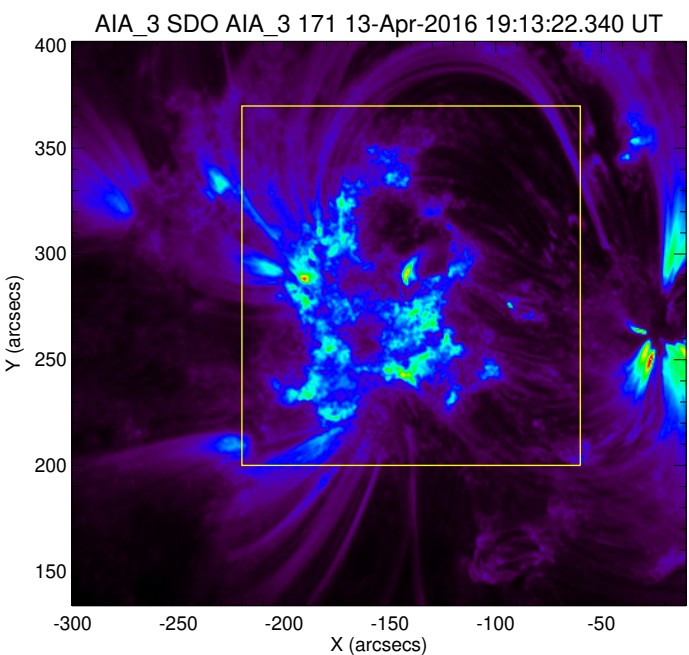

**Figure 1.** Intensity emission due to 171 Åwavelength of SDO/AIA at 19:13:22 UT. The green color shows the highest emission while the violet represents low emissiom. The yellow box is overlaid to show the region of interest (ROI) taken to analyse the flows at the footpoints of quiescent coronal loops.

In this paper, we study the quiescent coronal loops with big loop arches having one of their footpoints anchored at the edges for moss region. The different strands in such large loop systems have been identified using high resolution observations of SDO/AIA and studied the flows in it mapping the footpoints to the lower region of the solar atmosphere. The section 2 describes the observational data and its analyses presenting the details of the data used for our analyses. In Section 3, the results have been reported with their interpretations. In the last section, the discussions and conclusions are summarized.

## 2   Observational Data and Analyses

IRIS provides spectral data in the two UV domains: FUV band (1331.7 Å to 1358.4 Å and 1389.0 Å to 1407.0 Å) and NUV (2782.7 Å to 2835.1 Å) having a large number of spectral lines covering the photosphere, chromosphere, TR, and inner corona. Level 2 data is used for our study, which is calibrated for the dark current removal, flat fielding (De Pontieu et al. 2014). We have utilized Si IV (1393.78 Å), Mg II k (2796.20 Å), C II (1334.53 Å), and Ni I (2799.47 Å) spectral lines.





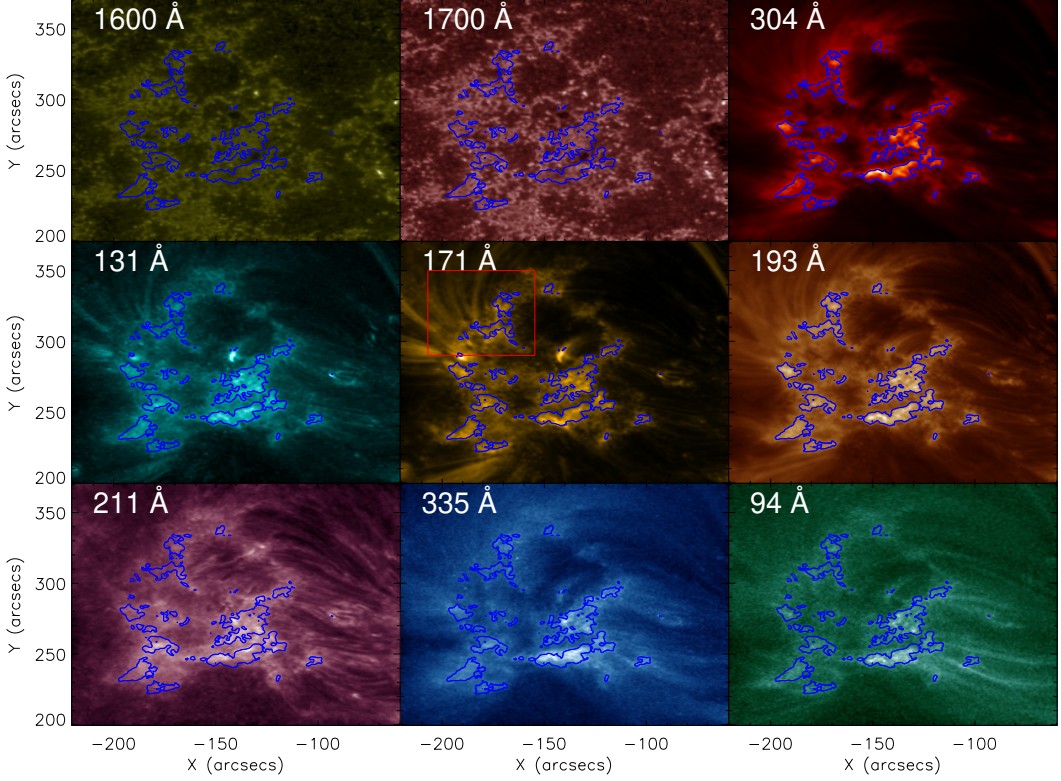

**Figure 2.** Mosaic representation of the zoom-in-view of the region of interest at different wavelength of SDO/AIA as mentioned on the corresponding panels.

In our present work, we have used dense raster data from IRIS for the time period 19:19:09 to 20:21:14 UTC on 2016 April 13 targeting the evolution of AR 12529 having slit width of $0.35''$ having step cadence of 9.3 s covering the field-of-view of $141''$ in x-direction and $175''$ in y-direction centered at the coordinates $(X_{\mathrm{cen}}, Y_{\mathrm{cen}}) = (-173'', 275'')$. The data is compensated for oscillations due to thermal variation using iris_orbitvarr_corr_l2.pro in the SSWIDL library. The rest wavelengths for different

5 spectral lines used in our analysis are calibrated using neutral lines from the relatively quiet-Sun area of the raster. The rest wavelength of Ni I used is 2944.4697 Å. Mg II k has been calibrated w.r.t. Ni I which is found to be 2796.3574 Å. Si IV line is calibrated w.r.t Fe I (1392.8052 Å) line and C II is calibrated w.r.t O I (1355.5987 Å). So, the calibrated wavelengths used for our analysis is 1393.7604 Å and 1334.5406 Å for Si IV and C II respectively.

Doppler velocities for different spectral lines Ni I; formation temperature: $\log(\mathrm{T\,/\,K}) = 4.2$, Mg II k (2796.20 Å; $\log(\mathrm{T\,/\,K}) =$

10 4.0), C II (1334.53 Å; $\log(\mathrm{T\,/\,K}) = 4.3$), and Si IV (1393.78 Å; $\log(\mathrm{T\,/\,K}) = 4.8$) have been calculated. The velocity resolution of IRIS is 1 km s$^{-1}$ (De Pontieu et al. 2014).

Si IV shows the characteristics of optically thin line and thus fitted with the single Gaussian while Ni I is absorption line and inverse Gaussian is fitted. Mg II k and C II are fitted with single or double Gaussian depending on their profile characteristics.





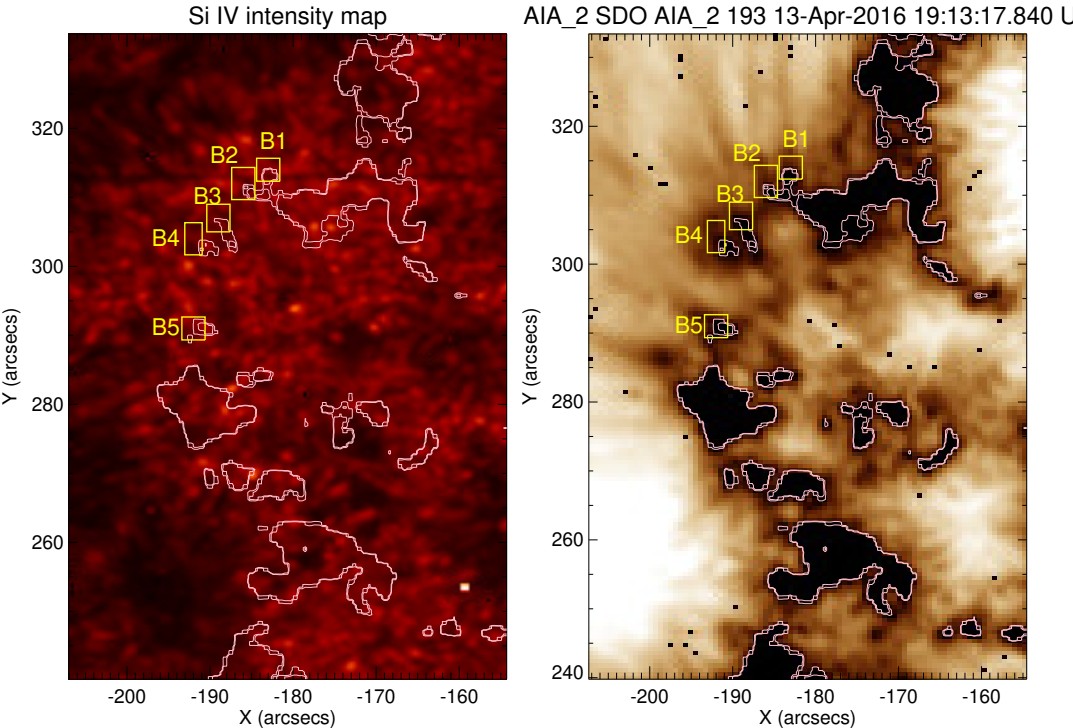

**Figure 3.** Left panel: Intensity map of Si IV (1393.78 Å) spectral line derived from the raster data of IRIS. Right panel: Identification of the footpoints of quiescent loops anchored at the moss regions. The different small boxes are taken at the footpoints of the individual loop strands.

The corresponding SDO/AIA observations are also taken in the different filters covering UV/EUV range corresponding to different temperature range in the solar atmosphere. AIA provides full-disk multi-wavelength observations of coronal lines having spatial resolution of 1.5" with a pixel size of 0.6" and temporal cadence of 12 s (Lemen et al. 2012).

## 3 Observational Results

Fig. 1 shows the intensity emission of AR 12529 having plage region and various loop arches anchored at 171 Å wavelength of SDO/AIA. The green emission predominantly indicates highest emission representing the moss region. The yellow box is overlaid to show the region of interest (ROI).

In Fig. 2, the moss region has been identified with the brightest emission in SDO/AIA 193 Å filter. The intensity threshold has been set which is shown by contours overlying on the different filters corresponding to different temperature ranges from the upper photosphere to corona. The northern segment of the moss regions where the quiescent coronal loops are anchored have been further taken to analyse the flows at the footpoints of these loop systems. The different bands of AIA show different





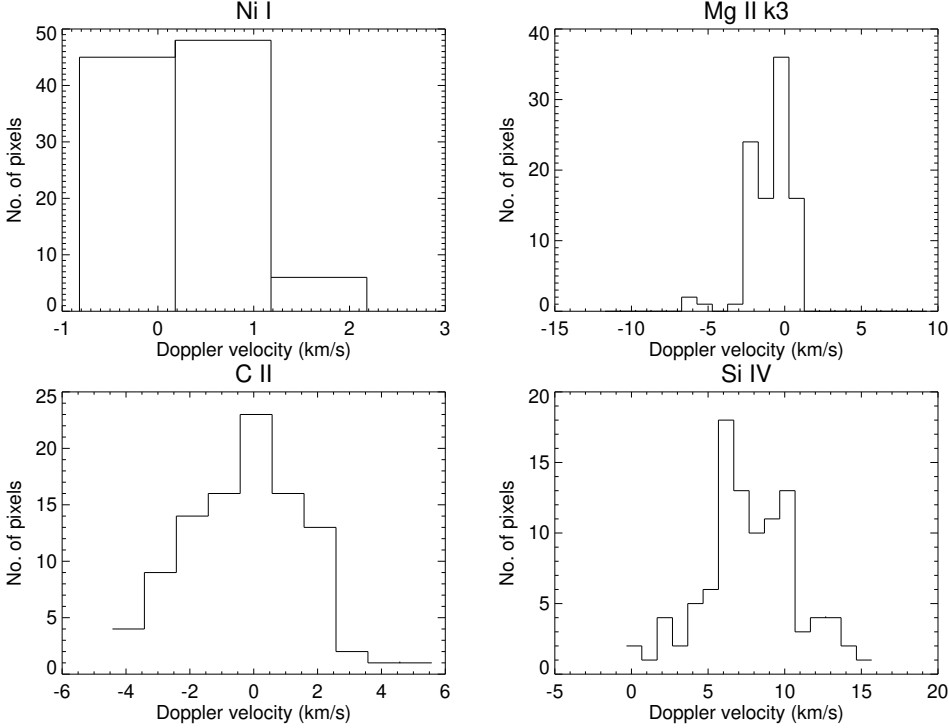

**Figure 4.** The velocity distributions for different ions corresponding to different temperatures at box B1.

morphological characteristics of an AR. 1600 Å and 1700 Å represent the continuum emission at the upper photosphere showing plage region near the active region. The 304 Å gives the chromospheric emission of the plage region. The hot loop structures are not distinctly visible. The middle row shows the inner coronal channels where quiescent coronal loops are clearly visible and moss region is identified. The last row shows the hot channels of SDO/AIA (335 Å, 211Å, and 94 Å) where the

5   quiescent loops taken for our analysis are not visible since quiescent loops are dominated by emissions from the temperatures ranging from 0.7 to 1 MK.

The left panel of Fig. 3 shows the intensity map of Si IV (1393.78 Å) line over which the boxes at the footpoints of the quiescent coronal loops are shown. The right panel of Fig. 3 is the emission of 193 Å line plotted in reverse color to identify the footpoints. Different boxes of different sizes are then chosen around the footpoints to cover the full strand of the loop. The

10  intensity map forming at TR temperature shows the plage region where moss region is same as surrounding plage.

The Doppler velocity distribution is thus explored at different location labelled as B1, B2, B3, B4, and B5. Positive values (red-shifts) represent downflows while the negative values (blue-shifts) indicate upflows. The Doppler velocity at each pixel in first box (B1) for different ions are then shown Fig. 4. The velocity distribution for Ni I shows the spread around 0 km/s ranging from $(-0.8$ to $+2.2)$ km s$^{-1}$. Mg II k shows the velocities ranging from $(-5$ to $+1)$ km s$^{-1}$ while C II has $(-5$ to $+5)$ km s$^{-1}$





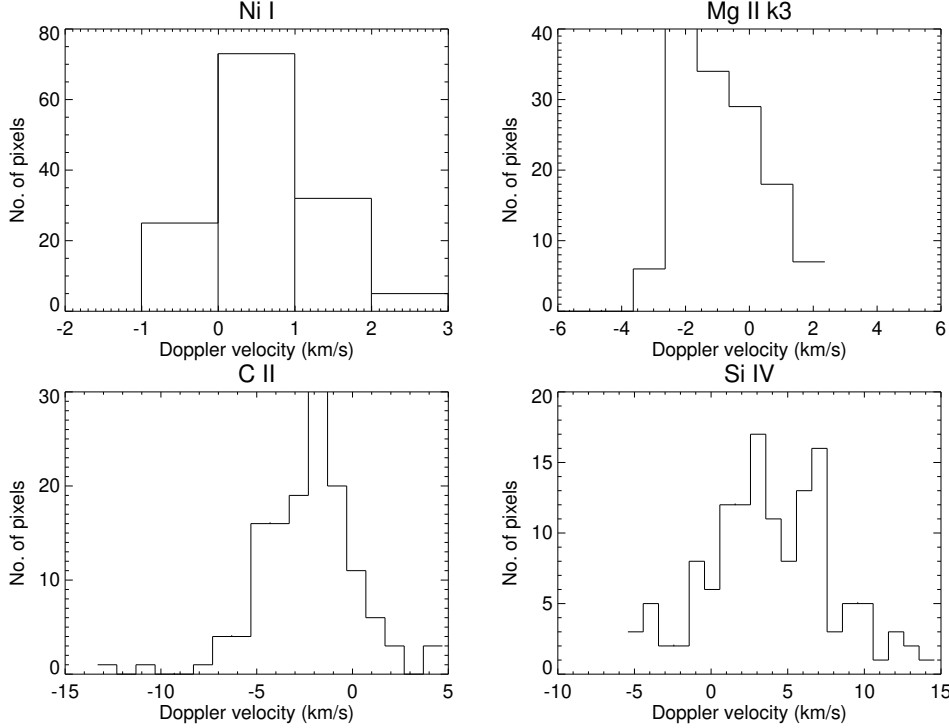

**Figure 5.** The velocity distributions for different ions corresponding to different temperatures at box B2.

. Si IV shows red-shifts having Doppler velocities ranging from $(0 \text{ to } +15)$ km s$^{-1}$ . The histogram of the Doppler velocity for different ions indicate the redshifts in the Si IV line and very small or negligible flows at Ni I, Mg II k, and C II.

Similarly, such Doppler velocity distribution is shown Fig. 5, Fig. 6, Fig. 7, and Fig. 8 for different boxes labelled as B2, B3, B4, and B5.

5   Fig. 9 shows the average Doppler shift of different ions as a function of their temperatures for different boxes chosen at the footpoints of the loops. The Doppler velocity of the Ni I line has negligible values indicating almost no flows $(0.27 \text{ to } 0.70)$ km s$^{-1}$ corresponding to photospheric region. The blueshifts (upflows) show small increment for B2, B4, and B4 $(-0.11 \text{ to } -0.31)$ km s$^{-1}$ while it remains almost same for B1 (0.16 km/s) and B5 $(0.80 \text{ km s}^{-1})$ up to the formation temperature of Mg II k. C II line shows considerable blueshifts (upflows) $(-0.17 \text{ to } -2.81)$ km s$^{-1}$ but it is still negligible as compared to

10   chromospheric flows. The Doppler velocity variation at Si IV shows prevalent redshifts (downflows) at all the locations corresponding to TR flows $(0.37 \text{ to } 6.97)$ km s$^{-1}$. The 1-sigmaa error is shown as error bars which is difficult to visualize in Fig. 9 owing to its very small values.



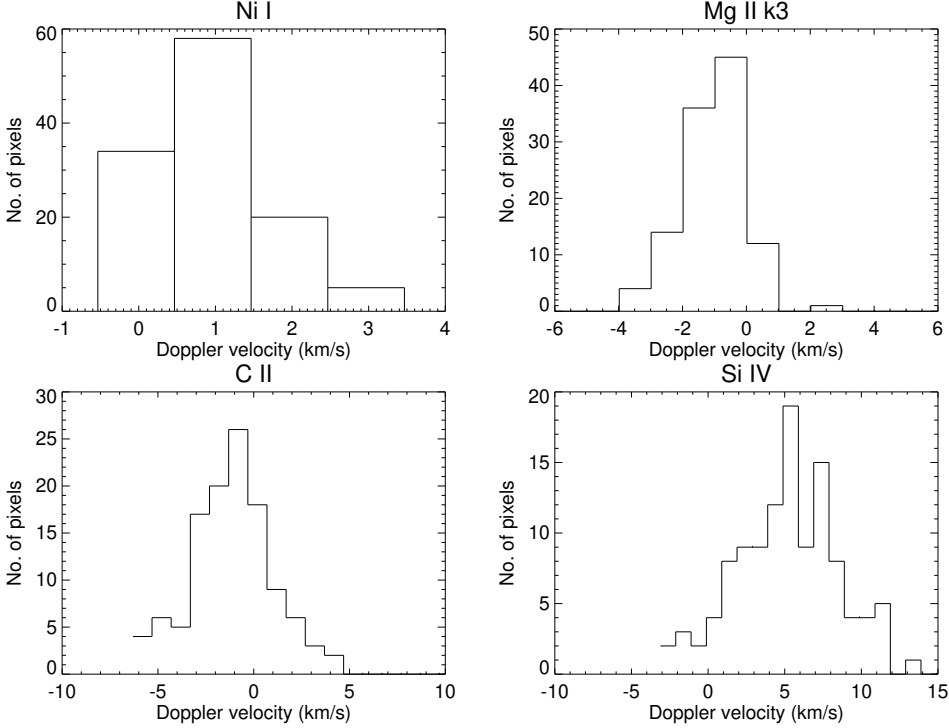

**Figure 6.** The velocity distributions for different ions corresponding to different temperatures at box B3.

# 4 Conclusions

We study the multi-spectral Doppler velocity variation at the footpoints of the quiescent coronal loops. The Doppler velocity variation shows red-shifts at the formation temperature of Si IV line corresponding to the TR.

Ni I, Mg II k, and C II shows very small velocities $(0.1 \text{ to } -2.81)$ km s$^{-1}$ corresponding to the photospheric as well as chromospheric region.

It has been previously shown that the moss regions show significant red-shifts (downflows) in the TR explaining the low-frequency heating (Bradshaw & Cargill 2010). The high- and low- frequency mechanisms depend upon the time taken by the loops to cool down as compared to heating frequency (Tripathi et al. 2008).

Our study of the flows at the quiescent coronal loops shows the similar characteristics as the dynamically active loops though the velocity values are less. The plasma predominantly shows redshifts at TR temperatures which corroborates with the low-frequency heating of loops in the coronal part of the solar atmosphere. These observations thus agree with the coronal loops heated up by low-frequency nanoflares via impulsive heating mechanism. These Doppler variation may also be caused by the asymmetries in the spectral profiles due to difference in the pressures (Mariska & Boris 1983). Though, asymmetry in





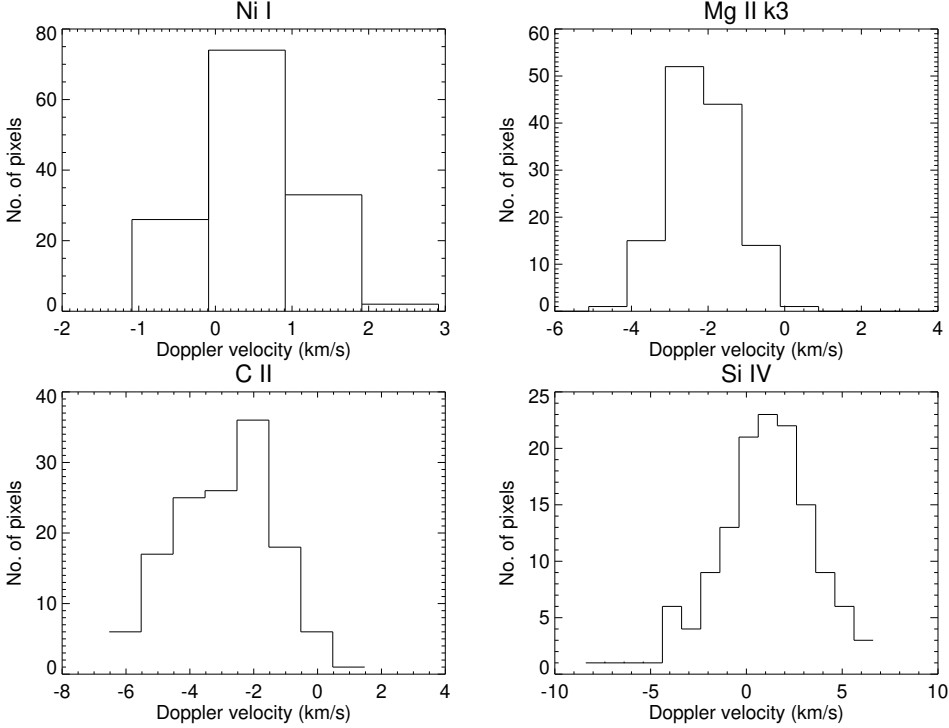

**Figure 7.** The velocity distributions for different ions corresponding to different temperatures at box B4.

the spectral profile has not been observed in our results, our measurements support the nano-flare driven impulsive heating mechanism even for the quiescent coronal loops which have steady flows.

*Acknowledgements.*  One of us (Yamini K. Rao) is fully supported by the financial grant from the ISRO RESPOND project. We acknowledge the use of IRIS observations. IRIS is a NASA small explorer mission developed and operated by Lockheed Martin Solar and Astrophysics
5  Laboratory (LMSAL) with mission operations executed at NASA Ames Research Center and major contributions to downlink communications funded by the Norwegian Space Center (NSC, Norway) through an ESA PRODEX contract.



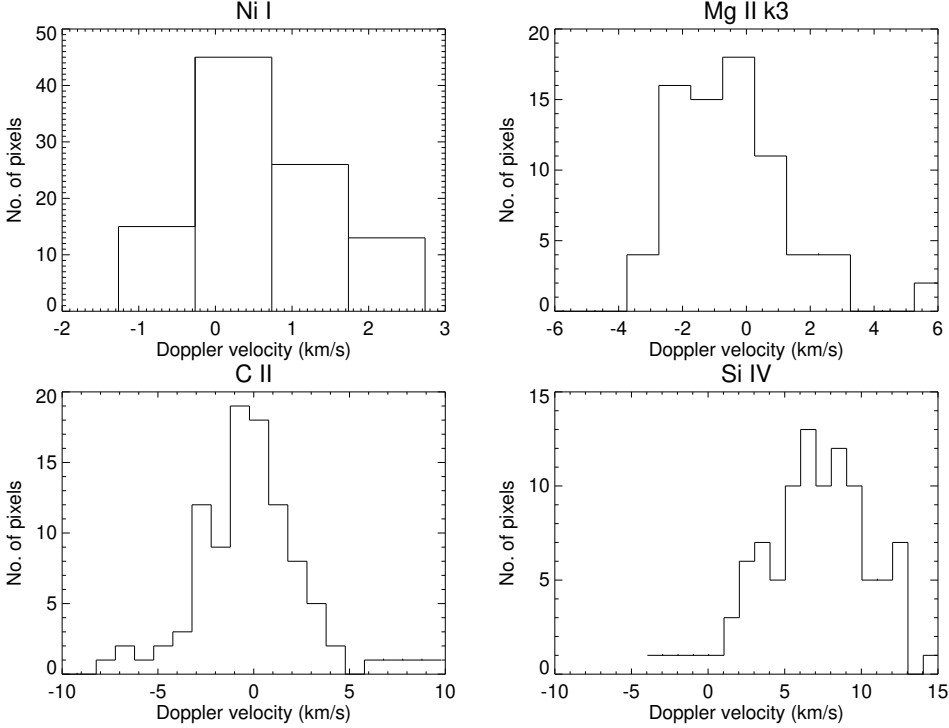

**Figure 8.** The velocity distributions for different ions corresponding to different temperatures at box B5.

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

De Pontieu, B.; Title, A. M.; Lemen, J. R.; Kushner, G. D.; Akin, D. J.; Allard, B.; Berger, T.; Boerner, P.; Cheung, M.; Chou, C.; Drake, J. F.; Duncan, D. W.; Freeland, S.; Heyman, G. F.; Hoffman, C.; Hurlburt, N. E.; Lindgren, R. W.; Mathur, D.; Rehse, R.; Sabolish, D. Seguin, R.; Schrijver, C. J.; Tarbell, T. D.; Wülser, J. -P.; Wolfson, C. J.; Yanari, C.; Mudge, J.; Nguyen-Phuc, N.; Timmons, R.; van Bezooijen, R.; Weingrod, I.; Brookner, R.; Butcher, G.; Dougherty, B.; Eder, J.; Knagenhjelm, V.; Larsen, S.; Mansir, D.; Phan, L.; Boyle, P.; Cheimets, P. N.; DeLuca, E. E.; Golub, L.; Gates, R.; Hertz, E.; McKillop, S.; Park, S.; Perry, T.; Podgorski, W. A.; Reeves, K.; Saar, S.; Testa, P.; Tian, H.; Weber, M.; Dunn, C.; Eccles, S.; Jaeggli, S. A.; Kankelborg, C. C.; Mashburn, K.; Pust, N.; Springer, L.; Carvalho, R.; Kleint, L.; Marmie, J.; Mazmanian, E.; Pereira, T. M. D.; Sawyer, S.; Strong, J.; Worden, S. P.; Carlsson, M.; Hansteen, V. H.; Leenaarts, J.; Wiesmann, M.; Aloise, J.; Chu, K. -C.; Bush, R. I.; Scherrer, P. H.; Brekke, P.; Martinez-Sykora, J.; Lites, B. W.; McIntosh, S. W.;




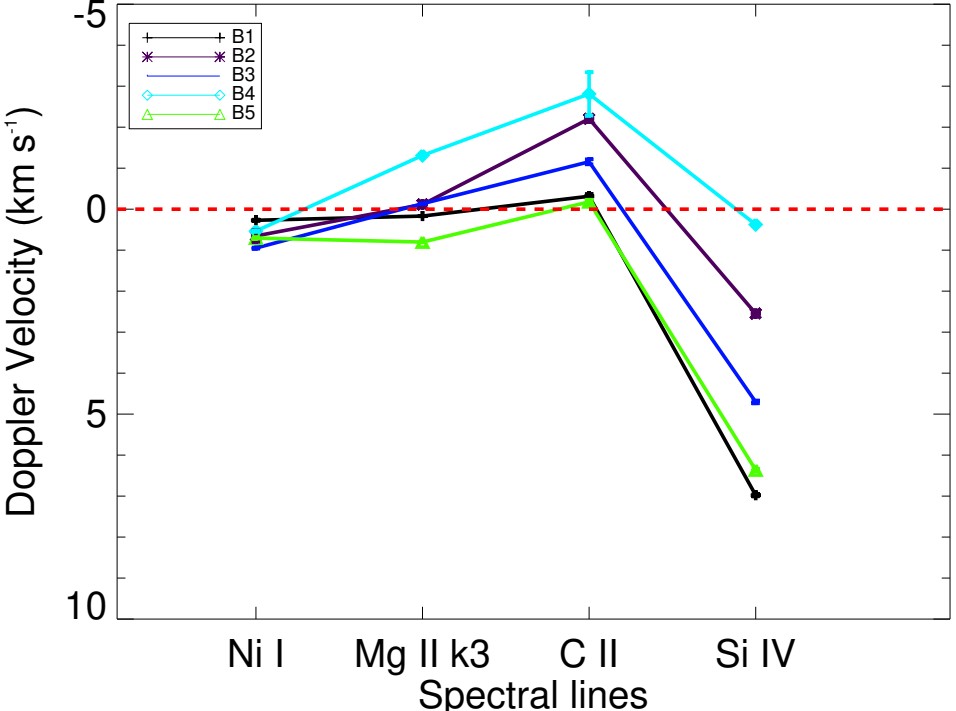

**Figure 9.** Average Doppler velocity variations for different ions dominating at different heights in the solar atmosphere for boxes B1, B2, b3, B4, and B5 at the footpoints of quiescent coronal loops.

Uitenbroek, H.; Okamoto, T. J.; Gummin, M. A.; Auker, G.; Jerram, P.; Pool, P.; Waltham, N., The Interface Region Imaging Spectrograph (IRIS), Solar Physics, 289, 2733, doi: 10.1007/s11207-014-0485-y , 2014

Del Zanna, G., & Mason, H. E.: Solar active regions: SOHO/CDS and TRACE observations of quiescent coronal loops A&A, 406, 1089, doi: 10.1051/0004-6361:20030791, 2003

5 Del Zanna, G.: Flows in active region loops observed by Hinode EIS , A&A, 481, L49, doi: 10.1051/0004-6361:20079087, 2008

Del Zanna, G., O'Dwyer, B., & Mason, H. E.: SDO AIA and Hinode EIS observations of "warm" loops, A&A, 535, A46, doi: 10.1051/0004-6361/201117470, 2011

Fletcher, L., & De Pontieu, B.: Plasma Diagnostics of Transition Region "Moss" using SOHO/CDS and TRACE, ApJL, 520, L135, doi: 10.1086/312157, 1999

10 Huang, Z., Xia, L., Li, B., & Madjarska, M. S.: Cool Transition Region Loops Observed by the Interface Region Imaging Spectrograph, Apj, 810, 46, doi: 10.1088/0004-637X/810/1/46, 2015

Klimchuk, J. A.: On Solving the Coronal Heating Problem, Solar Physics, 234, 41, doi: 10.1007/s11207-006-0055-z, 2006

Klimchuk, J. A.: Key aspects of coronal heating, Philosophical Transactions of the Royal Society of London Series A, 373, 20140256, doi: 10.1098/rsta.2014.0256, 2015



Lemen, J. R., Title, A. M., Akin, D. J.: The Atmospheric Imaging Assembly (AIA) on the Solar Dynamics Observatory (SDO), Solar Physics, 275, 17, doi: 10.1007/s11207-011-9776-8, 2012

Mariska, J. T., & Boris, J. P.: Dynamics and spectroscopy of asymmetrically heated coronal loops, ApJ, 267, 409, doi: 10.1086/160879,1983

Rao, Y. K., Srivastava, A. K., Kayshap, P., Wilhelm, K., & Dwivedi, B. N.: Plasma Flows in the Cool Loop Systems, ApJ, 874, 56, doi:
5      10.3847/1538-4357/ab06f5, 2019

Reale, F.: Coronal Loops: Observations and Modeling of Confined Plasma, Living Reviews in Solar Physics, 11, 4, doi: 10.12942/lrsp-2014-4, 2014

Tripathi, D., Mason, H. E., Young, P. R., & Del Zanna, G.: Density structure of an active region and associated moss using Hinode/EIS, A&A, 481, L53, doi: 10.1051/0004-6361:20079034 , 2008

10    Viall, N. M., & Klimchuk, J. A.: A Survey of Nanoflare Properties in Active Regions Observed with the Solar Dynamics Observatory, Apj, 842, 108, doi: 10.3847/1538-4357/aa7137, 2017

Warren, H. P., & Brooks, D. H.: The Temperature and Density Structure of the Solar Corona. I. Observations of the Quiet Sun with the EUV Imaging Spectrometer on Hinode, ApJ, 700, 762, doi: 10.1088/0004-637X/700/1/762 ,2009

Young, P. R., O'Dwyer, B., & Mason, H. E.:Velocity Measurements for a Solar Active Region Fan Loop from Hinode/EIS Observations,
15    ApJ, 744, 14, doi: 10.1088/0004-637X/744/1/14,2012