# Peer review of "Signatures of red-shifted footpoints in the quiescent coronal loop system"

_Annales Geophysicae, 2019_

## Referee Comment (RC1) · Anonymous Referee #1 · 17 Jun 2019

The work presented here deals with Doppler shifts in quiescent loops using observations from SDO and IRIS. There are some interesting results here, however, I believe that more work needs to be done to the paper before I think it can be published. The major concerns that I have at the minute are that: - The paper misses supplementary information in key places - It is not particularly clear the significance of the results - The discussion of the results is somewhat lacking - The writing style could be improved in places to make the paper more readable.

I will go through specific sections to highlight these issues in turn and will point out some questions that should be answered/considered prior to acceptance.

Introduction: You state: "Moss region being the subset of plage regions" What do you mean here? Note this could be a case of poor writing leading to confusion.

Observational Data and Analyses

Firstly, the heading of this section should be changed to something like "Observational Data". At present the heading is grammatically wrong, and you do not actually discuss any data analysis techniques here, only what the data was used and some small comments on how these images were prepared.

Secondly, you are missing key details here in regard to methods used. For example, how do you align IRIS and SDO images to each other? How accurate was this process? How do you deal with the varying image resolution in the two instruments? As you are using AIA images to determine the location of the moss to find the signatures in Doppler images with IRIS, you need to have the images accurately co-aligned. Presuming you've done that, you should therefore specify how you did and how successfully aligned the images are.

You use "w.r.t" frequently here. Please do not concatenate the phrase here and keep it as "with respect to" as "w.r.t" is not formal English.

You state: "Doppler velocities for different spectral lines Ni I; formation temperature: $\log(T / K) = 4.2$, Mg II k (2796.20 Å; $\log(T / K) = 10\ 4.0$), C II (1334.53 Å; $\log(T / K) = 4.3$), and Si IV (1393.78 Å; $\log(T / K) = 4.8$) have been calculated." Please consider re wording this to make it more legible. At the minute, the way this is presented is difficult to read meaningfully.

Observational Results

You state: "various loop arches anchored at 171 Å wavelength of SDO/AIA." How do you know that they are anchored at the height of 171? This seems like a rather speculative comment at present

You state: "predominantly indicates highest emission representing the moss region." I'm a little confused here, and by Fig 1 in terms of the color choices. The wording here should be different as "highest emission" does not specify anything in particular. I would

suggest using something like "highest intensity" as it is more specific, as "emission" could refer to something else.

Also, you state that the green colors represent moss. How can you know that for certain? To me at least it looks like you have put the image in a color table which shows up some potential moss in it based on intensity, but that doesn't necessarily mean that everything green is moss as other intensity enhancements will show up in the color table. For example, to the right and middle of the image you see green coloring in the base of a loop which does not appear to be associated with the moss in the boxed region. I imagine this intensity enhancement is due to some process at the bottom of this loop and not moss. Likewise, there is a similar case in the boxed region, so I would consider either rewording this bit or adding in the detected moss regions to the Figure to reinforce the point.

You state: "The intensity threshold has been set which is shown by contours overlying on the different filters". You do not at any stage explain how the intensity thresholding was performed. Please do that. This is important as it is an important step in how you isolate the moss regions. Also, more information allows the process to be repeatable by someone else.

How did you come to select the 5 regions that you looked at closer? At the moment there is very little to explain your selection criteria and why these regions were selected over another moss region.

You state: "are dominated by emissions from the temperatures ranging from 0.7 to 1 MK." Can you provide a citation for this temperature range to verify this statement?

You state: "The Doppler velocity of the Ni I line has negligible values indicating almost no flows (0.27 to 0.70) km s$-1$ corresponding to photospheric region." This statement is confusing and I don't know what you mean here. In general here, there is little discussion on the formation heights of the lines that you use, which has significance for how you interpret the doppler velocities you observe. I would suggest adding a bit

more on that as well within the text.

Conclusions

In general within the conclusions, I find that the discussion of the results is slightly lacking, and is not adequately covered in the preceding section. Traditionally conclusions are a section to make more general conclusions from the results you presented in previous sections. You should have stated these conclusions previously within the results and/or a discussion section, with the conclusions summarizing the key results. You discuss Doppler velocities in different lines in the previous section without much discussion on the significance of these values and their role in the conclusions that you come to.

You state: "These observations thus agree with the coronal loops heated up by low-frequency nanoflares via impulsive heating mechanism." How do you know that? You have not presented the evidence in such a way that confirms this conclusion. I would suggest going into more detail on this in the previous section, discussing your results in the context of other similar studies.

You state: "Though, asymmetry in the spectral profile has not been observed in our results". Can you prove that, and for all lines? I am doubtful that you have a purely symmetrical line profile across all lines for the duration of your observations. Therefore, there is a chance that this could have an effect and you should provide some evidence backing up your claim.

Also, please have a look at the grammar etc. in this paper very closely. I won't go into full details here as there are quite a few, but you should work to improve it. For example, a common problem you have is in the use of "the" throughout the text. In a lot of instances it is unnecessarily used e.g., "Our study of the flows at the quiescent coronal loops shows the similar characteristics as the dynamically active loops" would be better written as "Our study of flows in quiescent coronal loops display similar characteristics to dynamically active loops". In general with regards to "the", sometimes the overuse

of the word in the text effects the flow of the text and makes it more difficult to read. There are other instances in the text so please carefully consider the text in general from a grammatical perspective.

---

## Referee Comment (RC2) · Anonymous Referee #2 · 23 Jun 2019

The manuscript deals with the multiwavelength Doppler velocity estimates from space-borne instrumentation, for a quiet Sun region with coronal loop structures. The authors interpret the Doppler estimates as plasma flow patterns at loop footpoints and concludes the role of impulsive heating mechanism for the same. The paper looks interesting; however, some major concerns needs to be clarified before being considered for the publication. I will list the major science queries, followed by some minor corrections, related with grammar and else.

Major comments: (1). The spectral resolution for IRIS data corresponds to 1 km/s, as mentioned by the authors on Page 3, line:10. However, the Doppler estimates for Ni I wavelength are below this level and are totally unreliable. How these estimates can be used to infer the plasma flows in this passband? Same applies to estimates from Mg II

k and C II wavelengths as well.

(2). Page 3, line-13: Authors have acquired Doppler estimates by using single/double Gaussian fits to the line profiles. No such fits were shown. Please include the same, along with error estimates.

(3). Figure 3: SDO/AIA and IRIS intensity maps are shown with possible locations of loop footpoints. What is the photospheric magnetic field configuration at the footpoints and does it anyhow affect the plasma flows? Inclusion of an HMI LOS magnetogram for the same ROI would be useful.

(4). Doppler/FWHM maps for ROI should be included (maybe as a part of Fig. 3), to help the reader to get an idea of plasma flows at the loop footpoints and else.

(5). Page 7, last paragraph: The conclusion for plasma up/down flows is not clear. Authors have nowhere shown any signatures of either low-frequency heating or nano-flare heating. I am not sure how they have concluded the stated physical mechanisms for the analysed case. DEM analysis of the region can shed some light on impulsive heating in the loop structure.

Minor comments:

(1). Page 1, Line-20: The classification of the loops is based on their estimated thermal profile, or on the location/topology? Please clarify.

(2). Figure 1: Please add a colorbar to help the reader on the data range (highest, lowest emission).

(3). Page 2, Line-1: Rephrase the sentence "In this paper, we study . . . for moss region." It is very confusing now.

(4). Page 3, Line-4: "The rest wavelengths". What are rest wavelengths? Please clarify.

(5). Page 5, Last line (and else): Here, you have used the format km sˆ-1, while in

Figures 4-8, the format is km/s. Please be consistent and change accordingly.

(6). Page 6, Line-7: "The blueshifts (upflows) show small increment . . . chromospheric flows". Are the estimated increments below IRIS spectral resolution reliable? Please explain.

(7). Figures 4-9: The velocity distributions for "different ions". Here you are estimating Doppler shifts from wavelengths, observed from IRIS, and no ions were sampled for their Doppler shifts. Also, these ions emit at a range of temperatures, over a range of height in the solar atmosphere. Please rephrase to avoid confusion.

(8). Please check the grammar.

---

## Author Comment (AC1) · 24 Jul 2019

Introduction: You state: "Moss region being the subset of plage regions" What do you mean here? Note this could be a case of poor writing leading to confusion.

Reply: The sentence has been rephrased to "Moss is generally associated within the plage regions around the active regions sites." to make it more clear.

Observational Data and Analyses

Firstly, the heading of this section should be changed to something like "Observational Data". At present the heading is grammatically wrong, and you do not actually discuss any data analysis techniques here, only what the data was used and some small comments on how these images were prepared.

[Figure]

Reply: The heading has been changed to "Observational Data".

Secondly, you are missing key details here in regard to methods used. For example, how do you align IRIS and SDO images to each other? How accurate was this process? How do you deal with the varying image resolution in the two instruments? As you are using AIA images to determine the location of the moss to find the signatures in Doppler images with IRIS, you need to have the images accurately co-aligned. Presuming you've done that, you should therefore specify how you did and how successfully aligned the images are.

Reply: We have used the co-aligned Level2- data for which the the FOV matched IRIS 1400 SJI image has been cross-correlated with 1600 SDO/AIA filter almost of the same time. We are sure that our SDO/AIA images and IRIS SJI images are well correlated as per the above mentioned methodology. Plage regions are identified in SDO/AIA image data, and the location is mapped onto comparatively high-resolution IRIS data. However, we have derived the average Doppler velocity over the chosen moss regions (in various boxes) by deducing the integrated spectral line-profiles of various IRIS lines. Therefore, resolution mis-match in both the instruments should not be an issue in the present work. Our objective of the paper is to understand the behaviour of the bulk plasma flows over the chosen moss regions with respect to the formation temperature of various IRIS spectral lines. We have explained all these technical and scientific details in our revised paper carefully and correctly.

You use "w.r.t" frequently here. Please do not concatenate the phrase here and keep it as "with respect to" as "w.r.t" is not formal English.

Reply: It has been properly changed to "with respect to".

You state: "Doppler velocities for different spectral lines Ni I; formation temperature: $\log(T / K) = 4.2$, Mg II k (2796.20 Å; $\log(T / K) = 10$ 4.0), C II (1334.53 Å; $\log(T / K) = 4.3$), and Si IV (1393.78 Å; $\log(T / K) = 4.8$) have been calculated." Please consider rewording this to make it more legible. At the minute, the way this is presented is
difficult to read meaningfully.

Reply: Rephrased the sentecnce to "The Doppler velocities are deduced using different spectral lines, i.e., Ni I 2799.47 A, Mg II k 2796.20 A, C II 1334.53 A, and Si IV 1393.78 A respectively associated with the formation temperature of log(T / K) = 4.2, 4.0, 4.3, and 4.8."

Observational Results You state: "various loop arches anchored at 171 Å wavelength of SDO/AIA." How do you know that they are anchored at the height of 171? This seems like a rather speculative comment at present You state: "predominantly indicates highest emission representing the moss region." I'm a little confused here, and by Fig 1 in terms of the color choices. The wording here should be different as "highest emission" does not specify anything in particular. I would suggest using something like "highest intensity" as it is more specific, as "emission" could refer to something else.

Reply: Rephrased the sentence to "various loops anchored in the moss region are visible in 171 Å wavelength of SDO/AIA."

This sentence depicts that the loops are anchored in the plage/moss region, which are clearly visible in the 171 Å filter of SDO/AIA. Also, you state that the green colors represent moss. How can you know that for certain? To me at least it looks like you have put the image in a color table which shows up some potential moss in it based on intensity, but that doesn't necessarily mean that everything green is moss as other intensity enhancements will show up in the color table. For example, to the right and middle of the image you see green coloring in the base of a loop which does not appear to be associated with the moss in the boxed region. I imagine this intensity enhancement is due to some process at the bottom of this loop and not moss. Likewise, there is a similar case in the boxed region, so I would consider either rewording this bit or adding in the detected moss regions to the Figure to reinforce the point.

Reply: Fig. 1 shows the region-of-interest taken for our analysis. We have deleted the ambiguous text referring to green emission indicated by moss.

The colorbar has been added to show the intensity values where green color shows predominantly higher emission.

You state: "The intensity threshold has been set which is shown by contours overlying on the different filters". You do not at any stage explain how the intensity thresholding was performed. Please do that. This is important as it is an important step in how you isolate the moss regions. Also, more information allows the process to be repeatable by someone else.

Reply: The moss region has been identified using 193 A filter of SDO/AIA having intensity values double to that of surrounding plage region. The highest instensity shown by dark structures in the right panel of Figure 3 has been taken as moss. Our chosen criteria is now well described in the revised version of the paper in III (Observational Results) section.

How did you come to select the 5 regions that you looked at closer? At the moment there is very little to explain your selection criteria and why these regions were selected over another moss region.

Reply: The regions are chosen such that the moss region have loops associated to them. We have added this sentence to the paper.

The added sentence "The different boxes of different sizes are then chosen around the footpoints to cover the full strand of loop."

You state: "are dominated by emissions from the temperatures ranging from 0.7 to 1 MK." Can you provide a citation for this temperature range to verify this statement? You state: "The Doppler velocity of the Ni I line has negligible values indicating almost no flows (0.27 to 0.70) km s$-1$ corresponding to photospheric region." This statement is confusing and I don't know what you mean here. In general here, there is little discussion on the formation heights of the lines that you use, which has significance for how you interpret the doppler velocities you observe. I would suggest adding a bit

more on that as well within the text.

Reply: The emissions associated with 0.7 to 1 MK temperature correspond to 131, 171, and 193 A SDO/AIA lines. We have cited the paper Lemen et al., 2014 for your kind reference.

Ni I (2799.47 A) corresponds to upper photosphere, Mg II k gives emission ranging from mid-chromosphere to upper chromosphere. The core defined by (k3) forms ittle higher than the wings at 200 km below TR (Leenarts et al, 2013). C II core gives emission from 2.1 Mm while Si IV corresponds to the TR emission (Rathore et al., 2015).

The Doppler velocity values of Ni I has very small values (0.27 to 0.70) km s$-1$ which indicate almost no flows (i.e., no upflows or downflows near the photosphere). The values are reliable though to show that plasma is in steady state at the height of the formation temperature of Ni I and the neutral emissions come from the photospheric region. The co-spatial variation of Doppler velocities above the footpoints of the quiescent coronal loop systems at different heights correspond to formation temperatures of Ni I, Mg II k, C II, and Si IV.

Conclusions In general within the conclusions, I find that the discussion of the results is slightly lacking, and is not adequately covered in the preceding section. Traditionally conclusions are a section to make more general conclusions from the results you presented in previous sections. You should have stated these conclusions previously within the results and/or a discussion section, with the conclusions summarizing the key results. You discuss Doppler velocities in different lines in the previous section without much discussion on the significance of these values and their role in the conclusions that you come to.

Reply: The final section has been changed to "Discussions and Conclusions". The text has been added to make the discussions regarding red-shifts observed in the Si IV TR line in order to make the associated scientific descriptions more clear.

You state: "These observations thus agree with the coronal loops heated up by low-frequency nanoflares via impulsive heating mechanism." How do you know that? You have not presented the evidence in such a way that confirms this conclusion. I would suggest going into more detail on this in the previous section, discussing your results in the context of other similar studies.

Reply: Some ambiguity regarding the associated scientific context has been modified. The text has been changed accordingly. We do not claim to confirm the discussed possible physical scenario from the present observational base-line. However, we speculate the possibilities of most likely physical processes that may result into the response in form of the observed plasma flows in such loops. We have toned down the related sentences accordingly.

You state: "Though, asymmetry in the spectral profile has not been observed in our results". Can you prove that, and for all lines? I am doubtful that you have a purely symmetrical line profile across all lines for the duration of your observations. Therefore, there is a chance that this could have an effect and you should provide some evidence backing up your claim.

Reply: It is highly possible for asymmetries to be present in the profiles which supports impulsive heating mechanism. Also, the muli-thermal plasma is indicated by DEM maps at the footpoints of the quiescent coronal loop systems.

Also, please have a look at the grammar etc. in this paper very closely. I won't go into full details here as there are quite a few, but you should work to improve it. For example, a common problem you have is in the use of "the" throughout the text. In a lot of instances it is unnecessarily used e.g., "Our study of the flows at the quiescent coronal loops shows the similar characteristics as the dynamically active loops" would be better written as "Our study of flows in quiescent coronal loops display similar characteristics to dynamically active loops". In general with regards to "the", sometimes the overuse of the word in the text effects the flow of the text and makes it more difficult to read.

[Figure]

There are other instances in the text so please carefully consider the text in general from a grammatical perspective.

Reply: We have thoroughly read the paper and checked for grammatical mistakes. We have improved the text to make it easy to read.

Please also note the supplement to this comment:
https://www.ann-geophys-discuss.net/angeo-2019-66/angeo-2019-66-AC1-supplement.pdf

**Supplement:**

[revised manuscript text omitted]

---

## Author Comment (AC2) · 24 Jul 2019

(1). The spectral resolution for IRIS data corresponds to 1 km/s, as mentioned by the authors on Page 3, line:10. However, the Doppler estimates for Ni I wavelength are below this level and are totally unreliable. How these estimates can be used to infer the plasma flows in this passband? Same applies to estimates from Mg II k and C II wavelengths as well.

Reply: Though the upflows have values less than IRIS spectra resolution showing negligible or small plasma flows, the values are reliable enough within th error range. Our work emphasizes on the red-shift observed in TR line (Si IV).

(2). Page 3, line-13: Authors have acquired Doppler estimates by using single/double

[Figure]

Gaussian fits to the line profiles. No such fits were shown. Please include the same, along with error estimates.

Reply: The single Gussian fitting is done for optically thin lines while for the lines having multiple peaks, doble Gaussian fit has been done.

Mg II k line has been used w.r.t absorption core (also known as Mg II k3) which has been modelled by using two Gaussians (one positive and one negative). We have used straight line to fit the continuum.

(3). Figure 3: SDO/AIA and IRIS intensity maps are shown with possible locations of loop footpoints. What is the photospheric magnetic field configuration at the footpoints and does it anyhow affect the plasma flows? Inclusion of an HMI LOS magnetogram for the same ROI would be useful.

Reply: In our work, we were interested in showing the Doppler velocity trend at the footpoints of quiescent coronal loops with height. The HMI map has been shown in the revised paper where the whole plage region is dominated by postive magnetic polarities but the absence of any kind of mix polarities at the footpoints of the loop systems ruling out the possibilties of magnetic reconnection.

(4). Doppler/FWHM maps for ROI should be included (maybe as a part of Fig. 3), to help the reader to get an idea of plasma flows at the loop footpoints and else.

Reply: Intensity, Doppler velocity, FWHM map of Si IV and the related text has been included in the paper. The parametric plots are shown in Fig. 5.

(5). Page 7, last paragraph: The conclusion for plasma up/down flows is not clear. Authors have nowhere shown any signatures of either low-frequency heating or nano-flare heating. I am not sure how they have concluded the stated physical mechanisms for the analysed case. DEM analysis of the region can shed some light on impulsive heating in the loop structure.

Reply: It has been speculated that impulsive heating might be the cause of such flows

since there are symmetic as well as asymmetic profiles but still it does not rule out other possibilities. Also, the DEM analysis has been carried out and shown in Figure 4.

Minor comments: (1). Page 1, Line-20: The classification of the loops is based on their estimated thermal profile, or on the location/topology? Please clarify.

Reply: The loops have been classified on the basis of temperature range which has been included in the text as follows:

(2). Figure 1: Please add a colorbar to help the reader on the data range (highest, lowest emission).

Reply: The colorbar has been added.

(3). Page 2, Line-1: Rephrase the sentence "In this paper, we study . . . for moss region." It is very confusing now.

Reply: Rephrased to "In this paper, we study quiescent coronal loop arches having one of their footpoints anchored at the edges of moss region."

(4). Page 3, Line-4: "The rest wavelengths". What are rest wavelengths? Please clarify.

Reply: The rest wavelengths are the wavelengths which have been determined from the averaged profile of the quiet-Sun region ranging from -208.76" to -191.302" in x-direction and 205.248" to 221.8835" in y-direction of the raster for calibration purposes. The values of different spectral lines are mentioned in the revised paper.

(5). Page 5, Last line (and else): Here, you have used the format km sËĘ-1, while in Figures 4-8, the format is km/s. Please be consistent and change accordingly.

Reply: The notation has been made consistent to km sËĘ-1 throughout the paper.

(6). Page 6, Line-7: "The blueshifts (upflows) show small increment . . . chromospheric

flows". Are the estimated increments below IRIS spectral resolution reliable? Please explain.

Reply: The Doppler velocity values below IRIS resolution indicates the negligible or small plasma flows from upper photosphere to upper chromosphere. However, our work shows the Doppler velocity pattern where prominent red-shift has been observed in Si IV line. The small values have been used to indicate the Doppler velocity pattern at differnt heights.

(7). Figures 4-9: The velocity distributions for "different ions". Here you are estimating Doppler shifts from wavelengths, observed from IRIS, and no ions were sampled for their Doppler shifts. Also, these ions emit at a range of temperatures, over a range of height in the solar atmosphere. Please rephrase to avoid confusion.

Reply: The sentence has been changed to "different spectral lines" wherever "different ions" was mentioned.

(8). Please check the grammar.

Reply: It has been checked thoroughly.

Please also note the supplement to this comment:
https://www.ann-geophys-discuss.net/angeo-2019-66/angeo-2019-66-AC2-supplement.pdf

**Supplement:**

[revised manuscript text omitted]